# Vaccine Breakthrough Infections by SARS-CoV-2 Variants after ChAdOx1 nCoV-19 Vaccination in Healthcare Workers

**DOI:** 10.3390/vaccines10010054

**Published:** 2021-12-31

**Authors:** Pratibha Kale, Ekta Gupta, Chhagan Bihari, Niharika Patel, Sheetalnath Rooge, Amit Pandey, Meenu Bajpai, Vikas Khillan, Partha Chattopadhyay, Priti Devi, Ranjeet Maurya, Neha Jha, Priyanka Mehta, Manish Kumar, Pooja Sharma, Sheeba Saifi, Aparna Swaminathan, Sarfaraz Alam, Bharathram Uppili, Mohammed Faruq, Anurag Agrawal, Rajesh Pandey, Shiv Kumar Sarin

**Affiliations:** 1Department of Clinical Microbiology, Institute of Liver and Biliary Sciences, New Delhi 110070, India; drpratibhapgi@gmail.com (P.K.); niharikapatelmicro@gmail.com (N.P.); khillanv@yahoo.com (V.K.); 2Department of Clinical Virology, Institute of Liver and Biliary Sciences, New Delhi 110070, India; ektagaurisha@gmail.com (E.G.); sheetalnath.rooge@gmail.com (S.R.); amitpdy23@gmail.com (A.P.); 3Department of Hepatopathology, Institute of Liver and Biliary Sciences, New Delhi 110070, India; drcbsharma@gmail.com; 4Department of Transfusion Medicine, Institute of Liver and Biliary Sciences, New Delhi 110070, India; meenubajpaii@gmail.com; 5Council for Scientific and Industrial Research CSIR-Institute of Genomics and Integrative Biology (CSIR-IGIB), Mall Road, Delhi 110007, India; partha.c@igib.in (P.C.); priti.devi@igib.in (P.D.); ranjeet.maurya@igib.in (R.M.); neha.jha@igib.in (N.J.); priyanka.m@igib.in (P.M.); manish.dabas2705@gmail.com (M.K.); pooja.sharma@igib.in (P.S.); saifi.sheeba@yahoo.in (S.S.); aparnamurali10@gmail.com (A.S.); khan.sarfarazabbas@gmail.com (S.A.); bharathramh@gmail.com (B.U.); faruq.mohd@igib.in (M.F.); a.agrawal@igib.in (A.A.); rajeshp@igib.in (R.P.); 6Academy of Scientific and Innovative Research (AcSIR), CSIR-HRDC Campus, Ghaziabad 201002, India; 7Department of Hepatology, Institute of Liver and Biliary Sciences, New Delhi 110070, India

**Keywords:** vaccine breakthrough infections, COVID-19, healthcare workers, delta variant, immune response

## Abstract

This study elucidated the clinical, humoral immune response and genomic analysis of vaccine breakthrough (VBT) infections after ChAdOx1 nCoV-19/Covishield vaccine in healthcare workers (HCWs). Amongst 1858 HCWs, 1639 had received either two doses (1346) or a single dose (293) of ChAdOx1 nCoV-19 vaccine. SARS-CoV-2 IgG antibodies and neutralizing antibodies were measured in the vaccinated group and the development of SARS-CoV-2 infection was monitored.Forty-six RT-PCR positive samples from the 203 positive samples were subjected to whole genome sequencing (WGS). Of the 203 (10.92%) infected HCWs, 21.46% (47/219) were non-vaccinated, which was significantly more than 9.52% (156/1639) who were vaccinated and infection was higher in doctors and nurses. Unvaccinated HCWs had 1.57 times higher risk compared to partially vaccinated HCWs and 2.49 times higher risk than those who were fully vaccinated.The partially vaccinated were at higher risk than the fully vaccinated (RR 1.58). Antibody non-response was seen in 3.44% (4/116), low antibody levels in 15.51% (18/116) and medium levels were found in 81.03% (94/116). Fully vaccinated HCWs had a higher antibody response at day 42 than those who were partially vaccinated (8.96 + 4.00 vs. 7.17 + 3.82). Whole genome sequencing of 46 samples revealed that the Delta variant (B.1.617.2) was predominant (69.5%). HCWs who had received two doses of vaccine showed better protection from mild, moderate, or severe infection, with a higher humoral immune response than those who had received a single dose. The genomic analysis revealed the predominance of the Delta variant (B.1.617.2) in the VBT infections.

## 1. Introduction

Vaccines have emerged as an effective countermeasure against the accelerating global increase in severe acute respiratory syndrome coronavirus 2 (SARS-CoV-2) infection causing coronavirus disease 2019 (COVID-19). Multiple vaccine rollouts across the world will curb the pandemic by protecting vulnerable population groups [1].

In India, up until 30 April 2021, among 154 million cumulative vaccinations administered, 9.4 million healthcare workers (HCWs) had received a first dose and 6.2 million HCWs had received two doses [2]. There was a surge in COVID-19 casesfrom mid-March to the end of May 2021, with the worst affected states being Maharashtraand Delhi [3,4]. Similar surges in cases were reported from the UK, South Africa, and Brazil, with subsequent global spread [5]. The surge in India was predominantly associated with more transmissible variants with the potential for immune evasion.These variants included Alpha (B.1.1.7; 501Y.V1), Beta (B.1.351; 501Y.V2) and Gamma (B.1.1.28.1; 501Y.V3; P.1) [4]. India has reported the rapid spread of the Alpha variant and witnessed the emergence of a variant of concern, Delta (B.1.617.2) and a variant of interest, Kappa (B.1.617.1) [4]. For surveillance, the CDC has defined a vaccine breakthrough infection (VBT) as the detection of SARS-CoV-2 RNA or antigen in a respiratory specimen collected from a person ≥14 days after they have completed all recommended doses of a U.S. Food and Drug Administration (FDA)-authorized COVID-19 vaccine [5]. The USA has reported 0.01% VBT infections with the mRNA vaccines, while a chronic care medical facility in India has reported 16.8% VBT infections [5,6]. Initial vaccine efficacy trials for ChAdOx1 nCoV-19 showed that infections after vaccination occurred in 0.5% of participants. We studied a large cohort of vaccinated HCWs who had VBT infections, their clinical characteristics, immune response and genomic analysis of the causative SARS-CoV-2 virus.

## 2. Materials and Methods

The study was conducted at the Institute of Liver and Biliary Sciences (ILBS), Delhi, India in collaboration with the Council of Scientific and Industrial Research—Institute of Genomics and Integrative Biology (CSIR-IGIB). The study had the approval of the Institutional Ethical Committee (IEC/2020/77/MA07) and included the Declaration of Helsinki. The study included 1858 HCWs at our university hospital from 16 January 2021 to 31 May 2021. Vaccination was voluntary; 1639 HCWs received vaccination with ChAdOx1 nCoV-19 vaccine (two doses scheduled 28 days apart) and 219 declined vaccination. The HCWs were considered fully vaccinated after a period of 14 days following 2 doses of vaccine at least 28 days apart and amongst the vaccinated group, 1346 were fully vaccinated with a double dose of vaccine as at 31 May 2021. Amongst the HCWs, 293 had received the first dose by 31 March 2021, but had not taken the second dose as at 31 May 2021and were considered partially vaccinated at the end of the study period. Demographic details, including history of co-morbidities and prior history of COVID-19 infection were recorded. SARS-CoV-2 IgG antibodies and neutralizing antibodies were measured in the vaccinated group and monitored the development of SARS-CoV-2 infection. For the purposes of this study, we defined the VBT infections as the detection of SARS-CoV-2 RNA or antigen in a respiratory specimen collected from an individual who had received either one or two doses of ChAdOx1 nCoV-19 vaccine as this also had implications on the vaccine dose interval. The definitions of non-responders, reinfection and levels of antibodies are provided in the Appendix A.

### 2.1. Confirmation of Breakthrough Infection

Combined nasopharyngeal and oral swabs were collected in viral transport media (VTM) from all the vaccinated subjects at baseline before vaccination, on days 14 and 28 after the 1st dose of vaccine, and then on day 14 or later after the 2nd dose. The HCWs were instructed to report if they had any COVID-19 symptoms throughout the study period and were tested for SARS-CoV-2 infection with real time reverse transcriptase polymerase chain reaction (RT-PCR). Samples were tested for the presence of E and RdRP genes by RT-PCR using commercial kit (Q-Line^®^, M/s POCT Pvt Ltd., New Delhi, India) Samples with detection of both the genes with a < 35 cycle threshold (Ct) value were considered positive.

### 2.2. Humoral Immune Response

Blood samples were collected at baseline before vaccination, on days 14 and 28 after the 1st dose of vaccine, and then on day 14 or later after the 2nd dose in the vaccinated subjects. Samples were also collected 14 days post-infection from vaccinated subjects who were identified with infection. The SARS-CoV-2 IgG antibodies were measured using the enhanced chemiluminescence method (VitrosECi, Ortho Clinical Diagnostics, Raritan, NJ, USA). This is a qualitative assay based on a recombinant form of the spike subunit 1 protein. The results were based on the sample signal-to-cut-off (S/Co) ratio, with values <1.0 assessed as negative and ≥1.00 as positive results. Further the presence of neutralizing antibodies was measured bysurrogate neutralization ELISA in post-infection samples (GenScript Biotech, Piscataway, NJ, USA).

### 2.3. Whole Genome Sequencing

Sequencing using Illumina MiSeq Platform. Forty-six RT-PCR positive samples (Ct value ≤ 20) were randomly selected from the 203 positive samples from the virology repository at −80 °C and subjected to whole genome sequencing (WGS). WGS of the RNA elutes was carried out using the Genome Analyzer IIx (Illumina, San Diego, CA, USA) according to the manufacturer’s instructions. Double-stranded cDNA (ds cDNA) was synthesized from the RNA elutes. The first strand of cDNA was synthesized using the Superscript IV First strand synthesis system (Thermo Fisher Scientific, Waltham, MA, USA. Cat. No. 18091050) followed by single-stranded RNA (ssRNA) digestion with RNase H for second strand synthesis using DNA Polymerase I Large (Klenow) Fragment (New England Biolabs, Ipswich, MA, USA, Cat. No. M0210S). The ds cDNA was purified using AMPure XP beads (AMPure XP, Beckman Coulter, Brea, CA, USA, Cat. No. A63881) and quantified using NanoDrop (ND-1000 UV-Vis Spectrophotometer, Thermo Fisher Scientific). Then, 100 ng of purified ds cDNA was used for library prep using the Illumina DNA Prep with Enrichment kit (Illumina, Cat. No. 20018705). The process involves tagmentation followed by cleanup and amplification, leading to indexed DNA fragments. Following tagmentation and indexing, enrichment was performed using the Illumina RVOP (Illumina, Cat no. 20042472), wherein 500 ng of each sample was pooled by mass in accordance with the reference guide (Illumina, Doc. No. 1000000048041v05) for the 12-plex hybridization with biotinylated adjacent oligo-probes of the RVOP. The hybridization was performed overnight after which the probes were captured based on the streptavidin–biotin interactions. The final library was PCR amplified and purified before sequencing. The quality and quantity of the sequencing library was checked using Agilent 2100 Bioanalyzer with a high sensitivity DNA chip (Catalog number: 5067-4626) and the Qubit dsDNA HS Assay kit (Catalog number: Q32851), respectively. A loading concentration of 10pM was prepared by denaturing and diluting the libraries in accordance with the MiSeq System Denature and Dilute Libraries Guide (Illumina, Document no. 15039740 v10). Sequencing was performed on the MiSeq system, using the MiSeq Reagent Kit v3 (150 cycles) at 2 × 75 bps read length.

### 2.4. Sequencing Data Analysis

MiSeq data analysis. Fast QC v0.11.9 (http://www.bioinformatics.babraham.ac.uk/projects/fastqc accessed on 31 May 2021) was used to check the Phred quality score for all sequences. The quality score threshold was 20 and abovefor all samples. Adapter trimming was performed using the Trim Galore tool v0.6.1 (https://www.bioinformatics.babraham.ac.uk/projects/trim_galore/ accessed on 31 May 2021) and alignment of sequences was performed using the HISAT2 algorithm on human data build GRCh38 [7]. To remove any human sequences from the dataset, samtools v1.12 were used to remove the aligned sequences [8]. Henceforth, only unaligned sequences were taken into consideration. BCFTools v1.12 was used to generate consensus fasta and variant calling.

### 2.5. Phylogenetic and Mutation Analysis

We sequenced 46 vaccinated COVID-19 positive samples that were aligned to NC_045512 reference genome using the MAFFT v7.475 multiple alignment tool [9]. The aligned sequences were trimmed to remove gaps and a phylogenetic tree was generated using the default model of the IQ-TREE tool v2.0.3 [10]. The tree was visualized using FigTree v1.4.4 [11]. Further, the assembled genomes were assigned lineages using the package, Phylogenetic Assignment of Named Global Outbreak LINeages (PANGOLIN) [12]. The lollipop plot was generated in RStudio using g3viz, rtracklayer, and trackViewer packages followed by data visualization using the ggplot2 package. All the figures were updated using Inkscape software [13].

### 2.6. Data Availability

The consensusfasta generated for this study has been submitted in GISAID under the accessions: EPI_ISL_2424135 = 1, EPI_ISL_2426145 to EPI_ISL_2426189 = 45.

### 2.7. Data Analysis

The collected data were entered into an Excel spreadsheet and subsequently expressed as medians or percentages. The categorical data were analyzed using Chi-Square or Fisher’s exact test. The statistical analysis was performed using SPSS software version 22.

## 3. Results

### 3.1. Characteristics of Breakthrough Infections in HCWs

#### 3.1.1. SARS-CoV-2 Infections in HCWs

A total of 1858 HCWs were offered vaccination from 16 January 2021 to 31 May 2021; 1639 (88.2%) were vaccinated and 219 (11.7%) were non-vaccinated. (Table 1) Among the 1639 HCWs who were vaccinated, 1346 (82.1%) were fully vaccinated and 293 (17.9%) were considered partially vaccinated. Overall, SARS-CoV-2 infections were seen in 203 (10.9%) HCWs. Infections were more common in the non-vaccinated HCWs (47/219, 21.5%) than in those who were vaccinated (156/1639, 9.5%; *p* < 0.001) (Table 1). VBT infections were more common in partially vaccinated HCWs (40/293, 13.6%) than in fully vaccinated HCWs (116/1346, 8.6%; *p* = 0.008). The interval between vaccination dose and VBT infection was significantly longer in fully vaccinated HCWs (51 days, IQR 32–61) compared to those who were partially vaccinated (26 days, IQR 9–51.25, *p* = 0.007).

#### 3.1.2. Distribution of VBT Infections as Per Work Profile of HCW

The distribution of infections in non-vaccinated HCWs was as follows: doctors 27.7% (5/18), nurses 24.5% (27/110), technicians 20% (4/20), non-medical staff 16.6% (6/36) and 14.28% (5/35) in general duty assistants (GDA) (Table 1). The distribution of VBT infections in partially vaccinated HCWs was as follows: doctors 27.6%, nurses 24.1%, technicians 8.3%, non-medical staff 11.1%, and GDA 3.3%. Within the partially vaccinated group, significantly higher percentages of doctors and nurses had VBT infections compared to other HCW types (*p* < 0.001). The distribution of VBT infections in fully vaccinated HCWs was as follows: doctors 22%, nurses 24.2%, technicians 7.09%, non-medical staff 3.74% and GDA 1.22% (Table 1). Doctors and nurses in the fully vaccinated group were significantly more prone to acquire VBT as compared to technical staff, non-medical staff and GDA (*p* < 0.001).

#### 3.1.3. Reinfection

Reinfection was documented in two HCWs (doctors) in the fully vaccinated group (Ct values 33.1, 29.9), and two HCWs (nurse, non-medical staff) in the partially vaccinated group (Ct values 29.5, 20.5). All had mild symptoms of fever and body aches, but no significant lung involvement; none was immunocompromised.

#### 3.1.4. Clinical Presentation of VBT Infections

The median age was 34 (IQR: 21–67) years; 44.8% (91/203) were male and 55.2% (112/203) were female. The definitions of mild, moderate, and severe infection are given in the Appendix A. The majority of the infections were clinically mild, with fever, body aches, cough, headache, diarrhea, or vomiting, with a respiratory rate <24/min and SpO2 ≥94% in room air. The non-vaccinated subjects were at a significantly higher risk of developing infection as compared to partially (RR 1.57, (95% CI 1.07–2.31) (*p* = 0.02) and fully vaccinated HCWs (RR 2.49 (95% CI 1.83–3.39) *p* ≤ 0.001) (Table 2). Partially vaccinated HCWs were at a higher risk of developing infection compared to those who were fully vaccinated (RR 1.58 (95% CI, 1.13–2.22) (*p* = 0.01). No significant difference was noted in the risk of developing mild infection in the non-vaccinated compared to partially vaccinated HCWs (RR 1.0 (95% CI 0.6–1.65). The risk of developing moderate infection was significantly higher in unvaccinated than vaccinated HCWs (RR 3.35 (95% CI 1.5–7.45) (*p* = 0.002). None of the vaccinated subjects had severe infection requiring ICU admission and no death was reported. (Table 2). There were three (1.36%) severe cases and one death in the unvaccinated group.

#### 3.1.5. Change in IgG Antibody Levels from Baseline to 14 Days after the Second Dose

Baseline IgG levels were checked so that prior infection and its effect on post-vaccination immune responses could be considered. The IgG antibody levels at baseline (before vaccination) were negative (<1 signal/cut-off, S/CO) in 96.15% (150/156), while six HCWs had IgG antibodies at baseline. The antibody levels at day 14 and day 28 were also comparable in both of the groups (0.42 ± 1.18 vs. 0.67 ± 1.56, *p* = 0.37) and (3.78 ± 3.86 vs. 3.02 ± 2.67, *p* = 0.17), respectively (Figure 1). There was a significant difference in the partially vaccinated and fully vaccinated groups at 14 days after the second dose in the antibody response (7.17 ± 3.82 vs. 8.96 ± 4.00, *p* = 0.014) (Figure 1).The antibody levels were tested 14 days after the second dose and showed: non-response in 3.4% (4/116); low antibody levels (1–4.62 S/CO) in 15.5% (18/116); medium antibody levels (4.62–18.45 S/CO) in 81.0% (94/116); and no HCW had high levels of antibodies (>18.45 S/CO). The levels were defined as low, medium or high as previously described by Joyner et al. [14].

#### 3.1.6. Antibody Response Post-Infection

There were significant differences in IgG antibody and neutralizing antibody levels between fully and partially vaccinated HCWs 14 days post-infection. The IgG levels measured14 days post-infection were significantly higher in fully vaccinated HCWs (14.93 ± 5.76 vs. 12.89 ± 5.09, *p* = 0.05) (Mean ± SD). Similarly, the post-infection neutralizing antibody levels were significantly higher in fully vaccinated HCWs (57.28 ± 20.29 vs. 66.77 ± 23.04 (percentage inhibition), (*p* = 0.02).

#### 3.1.7. Genomic Sequencing of VBT Infections

Genomic sequencing and analysis were available for 46 samples. Thirty-two (69.6%) were from HCWs who had received two doses of vaccine and 14 (30.4%) were from those who had received a single dose of vaccine.

#### 3.1.8. Comparison of Ct Values

The approximate burden of viral load as measured indirectly by Ct value was similar in vaccinated and unvaccinated HCWs. The median Ct value was 21.1 (IQR 12.0–29.5) in partially vaccinated HCWs and 23.2 (IQR 0.0–33.1) in fully vaccinated HCWs (*p* = 0.82).

#### 3.1.9. Phylogenetic and Mutation Analysis

The B.1.617 lineage was found in 43 of 46 (93.5%) samples from vaccinated HCWs. (Figure 2). The B.1.617.2 (delta variant) (*n* = 32; 69.56%), B.1.617.1 (kappa variant) (*n* =11; 23.91%), and only one patient with B.1.1.7 strain (alpha variant) were identified (Figure 3). The presently widespread B.1.617.2 lineage-defining mutations were detected with significant frequency in our samples. The C23604G (S: P681R) mutation occurred in 95.65 percent of all sequences, while T22917G (S: L452R) and C22995A (S: T478K) mutations occurred in 93.48 percent and 63.04 percent, respectively (Figure 4). Other commonly occurring mutations included A23403G (S: D614G), G28881T (N:203), and G29402T (N:377). While the majority of these changes were non-synonymous, one synonymous mutation, C3073T, was found in all of the samples (Figure 4).

## 4. Discussion

Our prospective observational study details the clinical and genetic analysis of post-vaccination infections in HCWs, which coincided with a surge in SARS-CoV-2 infections during the second wave in Delhi. [15] A higher percentage of non-vaccinated HCWs were infected (21.5%) compared to vaccinated HCWs (9.5%). Infections were more common among partially vaccinated HCWs (13.6%) compared to fully vaccinated HCWs (8.6%), illustrating the benefit of two compared to one dose of vaccine. A study of HCWs at Christian Medical College, Vellore in India reported VBT infections in 9.6% of those fully vaccinated, 10.6% of those partially vaccinated, and 27.2% of those non-vaccinated [16]. Their cohort included HCWs who received either ChAdOx1 nCoV-19/Covishield or BBV152/Covaxin. Hall et al. reported 3.8% VBT infections in HCWs vaccinated with BNT162b2 mRNA vaccine (Pfizer) as compared to 38.5% in an unvaccinated cohort from England [17]. In a study from California, USA by Keehner et al., a positive rate of 0.05% was reported in HCWs after receipt of two doses of the BNT162b2 mRNA vaccine [18]. Benenson et al. reported VBT in 6.9% of vaccinated HCWs and 28.2% of unvaccinated subjects [19]. The data from three single-blind randomized controlled trials—one phase 1/2 study in the UK (COV001), one phase 2/3 study in the UK (COV002), and a phase 3 study in Brazil (COV003)—and one double-blind phase 1/2 study in South Africa (COV005) have shown that the overall vaccine efficacy 14 days after the second dose of the ChAdOx1 NCoV-19 vaccine was 66.7% [20]. Our study shows that fully vaccinated HCWs havea low risk of infection compared to partially and non-vaccinated groups. The risk of developing moderate illness requiring hospital admission was also lower in the fully vaccinated as compared to the other two groups.

The predominance of VBT among doctors and nurses as compared to the technicians, non-medical staff and GDA indicates higher risk amongst the immediate care givers owing to their exposure to infected patients and proximity during patient care. Thus, vaccination provides minimal protection against infection for HCWs with high levels of exposure. There was increased risk of infection amongst doctors and nurses compared to other hospital staff. Despite adequate use of personal protective equipment (PPE), HCWs who were directly involved in the care of COVID-19 patients were at increased risk. This highlights the significance of ensuring PPE quality and availability along with other aspects such as appropriate use, regular training for donning and doffing of PPE, and the clinical situation and procedures involved. Consequently, the protective effect of vaccination seems to reflect the protection offered to HCWs with lower levels of exposure within the healthcare setting. Another interesting observation from the study was the time to infection after the second dose, which occurred during the time period when antibody titers should be at their peak. Thus, at the time when the vaccines were supposed to be working the best, protection against developing infection was minimal. However, as no severe cases were seen in the vaccinated group, thus vaccination provided protection against severe disease. There is no published data for VBT infections in different categories of HCWs.

Higher severity of infection was also reported in the non-vaccinated group; three HCWs were admitted to the ICU and there was one death. Clinically, mild infection with fever and sore throat was the most common presentation in both vaccinated and non-vaccinated HCWs. Similar clinical findings were also documented by Philomina et al. in breakthrough infections in six people receiving two doses of ChAdOx1 nCoV-19/Covishield vaccine in Kerela [21]. In a recent study of breakthrough infections from Delhi, including a mixed cohort of HCWs, 10 received ChAdOx1 nCoV-19/Covishield vaccine and 53 received BBV152/Covaxin and none had severe infection [22]. Thus, two doses of ChAdOx1 nCoV-19/Covishield vaccine offer protection against moderate to severe COVID-19 disease.

The serological analysis showed significantly higher antibody response in fully vaccinated HCWs as compared to partially vaccinated ones. Single-dose recipients had higher VBT infection rate and lower humoral response, predisposing them to moderate to severe clinical disease.

There was a predominance of B.1.617 lineage in a phylogenetic examination of 46 vaccinated COVID-19 positive samples collected in the months of April and May 2021, which coincided with a massive surge of cases in Delhi. The Delta variant was predominant in nearly 70% of the samples. As B.1.617.2 is dominating the numbers here, this suggests that there is a higher vaccine breakthrough risk with B.1.617.2 as compared to B.1.617.1 and B.1.1.7. It is known that B.1.617.2 is characterized by 3 spike mutations L452R, T478K and P681R, and B.1.617.1 is characterized by L452R, E484Q and P681R [23]. A similar predominance of B.1.617.2 in vaccinated individuals was also reported from another tertiary care center in Delhi during the surge of infections in March and April [6]. Data from the Indian SARS-CoV-2 Genomic Sequencing Consortium (INSACOG) also show the predominance of B.1.617.2 since April in the general population in Delhi [24].

The variant of concern, B.1.617.1 partially impairs neutralizing antibodies elicited by BNT162b2 vaccine [25]. Similar evasion from ChAdOx1 nCoV-19 (Covishield) vaccine induced antibodies is possible by the B.1.617.2 variant. The RBD mutation, T478K is unique to B.1.617.2 and may facilitate antibody escape [26].

Genetic variants are associated with immune escape mechanisms (immune escape variants). Despite the existence of several RBD mutations, studies to examine the capability of vaccinee sera to neutralize circulating variants showed that strains such as B.1.1.7 remain potently neutralized. Several RBD-specific antibodies can bind only the open spike protein and it has been discovered that D614G renders the spike protein more sensitive to neutralizing antibodies by increasing the likelihood of the open conformation occurring [25,26]. Nevertheless, other circulating variations escape vaccine-induced humoral immunity [27]. The L452R mutation was found in roughly 93 percent of the vaccinated COVID-19 positive samples. Leucine-452 is located in the RBD receptor-binding motif, at the point of direct interaction with the ACE2 receptor. Its substitution with arginine is expected to result in substantially greater receptor affinity as well as escape from neutralizing antibodies [28]. The structural investigation of RBD mutations L452R and E484Q, as well as P681R at the furin cleavage region, revealed the possibility of enhanced ACE2 binding and an increased rate of S1–S2 cleavage, resulting in improved transmissibility. The same two RBD mutations resulted in reduced binding to selected monoclonal antibodies (mAbs), thus decreasing neutralizing capacityof the mAbs, according to an Indian study [28].

There are certain limitations of the study. Genomic analysis was restricted to breakthrough infections, positive samples from unvaccinated group were not sequenced. The data represented the VBT infections in hospital settings where the exposure to infection was higher and may not reflect the infection rates in the general population. The study findings could not differentiate between the relative contribution of hospital versus community acquisition of infection for the HCWs. The study did not address the difference in risk behaviors for those fully, partially, and non-vaccinated, and the possible association with the initial decision to be vaccinated and subsequent lack of COVID-19 appropriate behavior. Testing of antibodies with quantitative assay and performing tests using different dilutions of samples could not be done due to resource constraint. Further immune surveillance and characterization of vaccine breakthrough infections and associated variants is warranted to further understand disease pathophysiology and to effectively design measures to curb the spread of SARS-CoV-2.

## 5. Conclusions

HCWs who had received two doses of vaccine showed better protection from mild, moderate, or severe infection, with a higher humoral immune response than those who had received a single dose. The genomic analysis revealed the predominance of the Delta variant (B.1.617.2) in the VBT infections.

## Figures and Tables

**Figure 1 vaccines-10-00054-f001:**
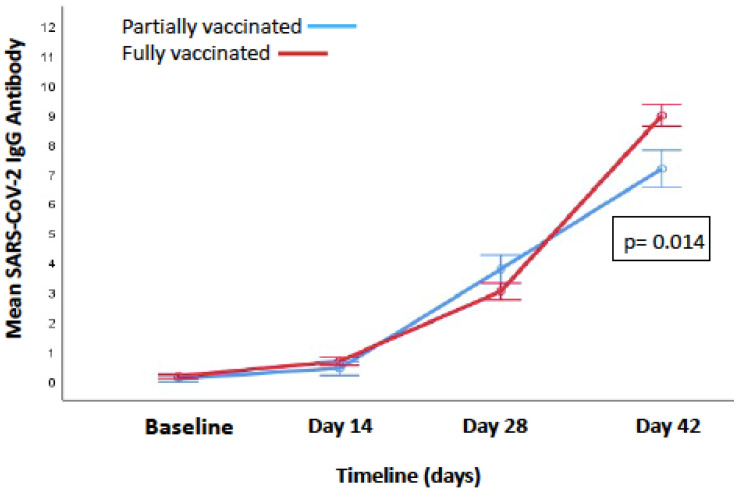
Difference in the SARS-CoV-2 IgG antibody levels from baseline to day 42 in partially and fully vaccinated HCWs.

**Figure 2 vaccines-10-00054-f002:**
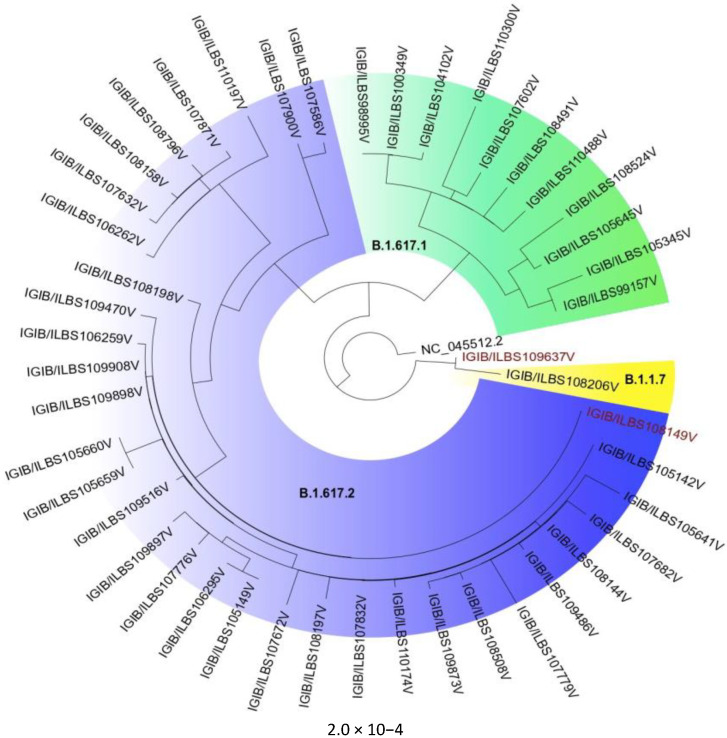
The phylogenetic distribution of lineages in 46 vaccinated samples.

**Figure 3 vaccines-10-00054-f003:**
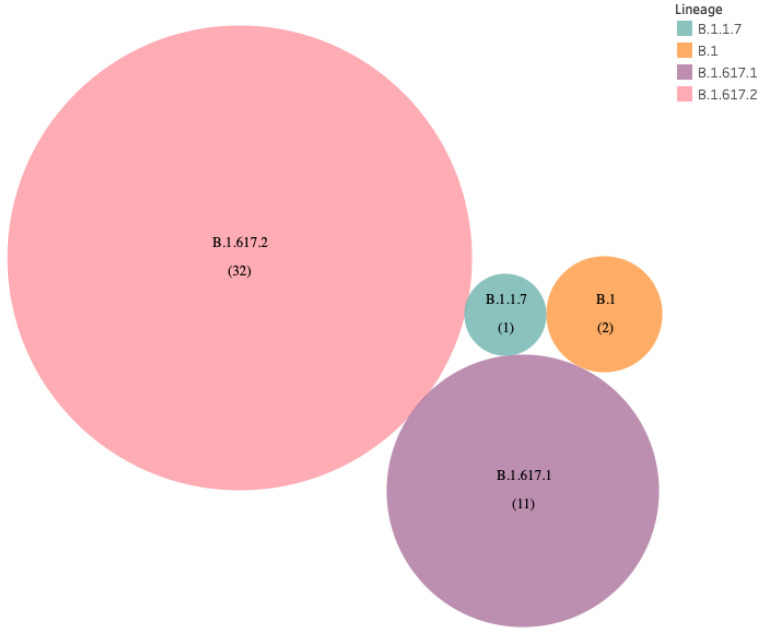
Prevalence of B.1.617.2 variants in post-vaccinated samples.

**Figure 4 vaccines-10-00054-f004:**
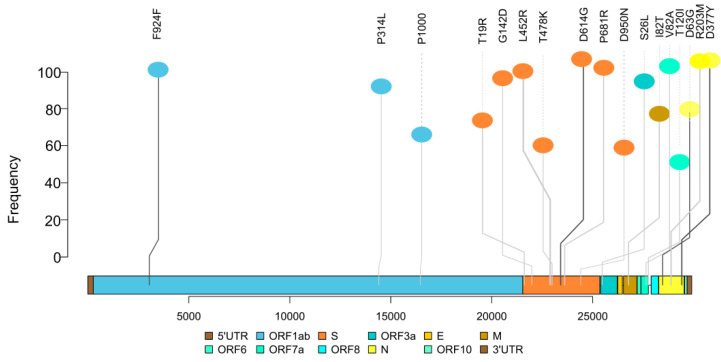
Top most frequent mutations detected in 46 vaccine-breakthrough samples.

**Table 1 vaccines-10-00054-t001:** Distribution of VBT infections as per work profile of HCW.

Category of HCW	Vaccinated *N* = 1639	Partially Vaccinated *N* = 293	Fully Vaccinated *N* = 1346	Non-Vaccinated *N* = 219	Total *N* = 1858	Level of Significance *p* Value
Doctors *n*/N (%)	30/129 (23.26)	8/29 (27.59)	22/100 (22)	5/18 (27.78)	35/147 (23.81)	0.754
Nursing staff *n*/N (%)	76/314 (24.20)	21/87 (24.14)	55/227 (24.23)	27/110 (24.55)	103/424 (24.29)	0.997
Technicians *n*/N (%)	16//215 (7.44)	5/60 (8.33)	11/155 (7.10)	4/20 (20)	20/235 (8.51)	0.150
Non-Medical *n*/N (%)	14/321 (4.36)	3/27 (11.11)	11/294 (3.74)	6/36 (16.67)	20/357 (5.60)	0.003 *
GDA *n*/N (%)	20/660 (3.03)	3/90 (3.33)	17/570 (2.98)	5/35 (14.29)	25/695 (3.60)	0.002 *
Total *n*/N (%)	156/1639 (9.52)	40/293 (13.65)	116/1346 (8.62)	47/219 (21.46)	203/1858 (10.93)	non-vaccinated vs. vaccinated <0.001 Partially vaccinated vs. fully vaccinated *p* = 0.008
*p* value		<0.001 ^#^	<0.001 ^#^	0.621	<0.001 ^#^	

N = total number subjects in the category, *n* = number of subjects with infection in the category, GDA: General duty assistant and housekeeping staff. * = fully vaccinated vs. non-vaccinated and partially vaccinated vs. non-vaccinated have significant difference. No significant difference between fully vaccinated and partially vaccinated subjects. ^#^ = Doctors, nursing staff vs. technical staff, non-medical staff and GDA have significant difference.

**Table 2 vaccines-10-00054-t002:** Clinical presentation of VBT infections in HCWs and association with vaccination.

Infection Status	Non-Vaccinated (*n* = 219)	Partially Vaccinated (*n* = 293)	RR ^a^ Non-Vaccinated vs. Partially Vaccinated	Fully Vaccinated (*n* = 1346)	RR ^b^ Non-Vaccinated vs. Fully Vaccinated	RR ^c^ Partially Vaccinated vs. Fully Vaccinated
Developed infection N (95% CI)	47 (21.46%)	40 (13.65%)	1.57 (1.07–2.31) *p* = 0.02	116 (8.61%)	2.49 (1.83–3.39) *p* ≤ 0.001	1.58 (1.13–2.22) *p* = 0.01
Mild infection	24 (10.9%)	32 (10.9%)	1.0 (0.6–1.65) *P* = 0.99	113 (8.39%)	1.30 (0.86–1.98) *p* = 0.21	1.30 (0.9–1.8) *p* = 0.17
Moderate illness requiring Hospitalization	20 (9.13%)	8 (2.73%)	3.35 (1.5–7.45) *p* = 0.002	3 (0.22%)	40.97 * (12.2–136.7) *p* < 0.001	12.25 * (3.2–45.9) *p* < 0.001
ICU admission	3 (1.36%)	0 (0)	-	0 (0)	-	-
Death	1 (0.46%)	0 (0)		0 (0)	-	-

RR—Relative risk, ICU—Intensive care unit, HCW—healthcare worker. ^a^·Comparison of the risk of getting infection in nonvaccinated group with the partially vaccinated group. ^b^·Comparison of the risk of getting infection in nonvaccinated group with the fully vaccinated group. ^c^·Comparison of the risk of getting infection in the partially vaccinated group with the fully vaccinated group. * RR Values are high as the number of infections in the vaccinated groups are very small.

## Data Availability

The consensus fasta generated for this study has been submitted in GISAID under the accessions: EPI_ISL_2424135 = 1, EPI_ISL_2426145 to EPI_ISL_2426189 = 45.

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
