# Peer review of "Vaccine Breakthrough Infections by SARS-CoV-2 Variants after ChAdOx1 nCoV-19 Vaccination in Healthcare Workers"

_vaccines, 2021, doi:10.3390/vaccines10010054_

Round 1
Reviewer 1 Report
In introduction some sentences need citation, e.g. lines 44 and 45 sentence: “Similar surges in cases were reported from the UK, South Africa, and Brazil, with subsequent global spread.”
Table1 and 2 can be merged according to the variables and divided according to the subgroups compared, to better present the results for readers.
Quality of images are too low and shall be improved.
Decision: Accept, Minor essential revisions
Author Response
Point 1: In introduction some sentences need citation, e.g. lines 44 and 45 sentence: “Similar surges in cases were reported from the UK, South Africa, and Brazil, with subsequent global spread.”
Response 1: References have been added to the lines 44, 45,53.
Point 2: Table1 and 2 can be merged according to the variables and divided according to the subgroups compared, to better present the results for readers.
Response 2: We have tried to merge the tables, but the variables are different so cannot combine the revelant information in one table, hence requesting you to consider the two tables separate.
Point 3: Quality of images are too low and shall be improved.
Response 3: The previous images have been changed with improved resolution. (Figure 2, 3,4)
Reviewer 2 Report
Nice study.
Table 1 - please comment on the apparent difference between doctors/nurses and other staff, perhaps add see discussion.
Lines 208-218 - could you present the antibody levels as histograms (perhaps in an appendix) - just to ensure that the distributions are not skewed......same for section 3.1.5
Section 3.1.6 are these proportions significantly different?
Do you have enough data to do analysis of frequency of the types of symptoms in the different groups?
It would also be interesting to see frequency of symptoms and severity based on strength of the humoral response.
The general advice is to dig a bit more deeply into the available data - too good to waste such an opportunity.
Author Response
Point 1: Table 1 - please comment on the apparent difference between doctors/nurses and other staff, perhaps add see discussion.
Response 1: There was increased risk of infection amongst doctors and nurses compared to other hospital staff. Despite adequate use of personal protective equipment (PPE), HCWs who were directly involved in care of COVID-19 patients were at increased risk. This highlights the significance of ensuring PPE quality and availability along with other aspects like appropriate use, regular training for donning and doffing of PPE, and clinical situation and procedures involved.
The above comment added to line 289- 293
Point 2: Lines 208-218 - could you present the antibody levels as histograms (perhaps in an appendix) - just to ensure that the distributions are not skewed......same for section 3.1.5
Response 2:.Histograms are now provided in the appendix section. These histograms are for the IgG antibody distribution at 42 days (14 days after second dose). The histograms for post infection antibody response are given in appendix.
Point 3: Section 3.1.6 are these proportions significantly different?
Response 3: Genomic sequencing was done for random 46 samples with Ct values less than 21. Hence we cannot assign significance to the difference in both groups as both are mutually exclusive. It’s a chance finding that majority of the randomly picked samples are from fully vaccinated individuals. Thus we cannot comment that these proportions are significantly different.
Point 4: Do you have enough data to do analysis of frequency of the types of symptoms in the different groups?
It would also be interesting to see frequency of symptoms and severity based on strength of the humoral response.
Response 4: We had collected the data on the types of symptoms in the breakthrough infections, however all were respiratory infections with similar presentation. Hence we did not discuss about the difference of the symptoms. The difference in severity of the infections has been elaborated in the table 2.
Round 2
Reviewer 2 Report
Thank you for the changes.